# Leveraging Sparsity for Efficient Submodular Data Summarization

**Erik M. Lindgren,  Shanshan Wu,  Alexandros G. Dimakis**
The University of Texas at Austin
Department of Electrical and Computer Engineering
erikml@utexas.edu, shanshan@utexas.edu, dimakis@austin.utexas.edu

## Abstract

The facility location problem is widely used for summarizing large datasets and has additional applications in sensor placement, image retrieval, and clustering. One difficulty of this problem is that submodular optimization algorithms require the calculation of pairwise benefits for all items in the dataset. This is infeasible for large problems, so recent work proposed to only calculate nearest neighbor benefits. One limitation is that several strong assumptions were invoked to obtain provable approximation guarantees. In this paper we establish that these extra assumptions are not necessary—solving the sparsified problem will be almost optimal under the standard assumptions of the problem. We then analyze a different method of sparsification that is a better model for methods such as Locality Sensitive Hashing to accelerate the nearest neighbor computations and extend the use of the problem to a broader family of similarities. We validate our approach by demonstrating that it rapidly generates interpretable summaries.

## 1   Introduction

In this paper we study the facility location problem: we are given sets $V$ of size $n$, $I$ of size $m$ and a *benefit* matrix of nonnegative numbers $C \in \mathbb{R}^{I \times V}$, where $C_{iv}$ describes the benefit that element $i$ receives from element $v$. Our goal is to select a small set $A$ of $k$ columns in this matrix. Once we have chosen $A$, element $i$ will get a benefit equal to the best choice out of the available columns, $\max_{v \in A} C_{iv}$. The total reward is the sum of the row rewards, so the optimal choice of columns is the solution of:

$$\underset{\{A \subseteq V : |A| \le k\}}{\arg \max} \sum_{i \in I} \max_{v \in A} C_{iv}. \tag{1}$$

A natural application of this problem is in finding a small set of representative images in a big dataset, where $C_{iv}$ represents the similarity between images $i$ and $v$. The problem is to select $k$ images that provide a good *coverage* of the full dataset, since each one has a close representative in the chosen set.

Throughout this paper we follow the nomenclature common to the submodular optimization for machine learning literature. This problem is also known as the maximization version of the $k$-medians problem or the submodular facility location problem. A number of recent works have used this problem for selecting subsets of documents or images from a larger corpus [27, 39], to identify locations to monitor in order to quickly identify important events in sensor or blog networks [24, 26], as well as clustering applications [23, 34].

We can naturally interpret Problem 1 as a maximization of a *set function* $F(A)$ which takes as an input the selected set of columns and returns the total reward of that set. Formally, let $F(\emptyset) = 0$ and

for all other sets $A \subseteq V$ define

$$F(A) = \sum_{i \in I} \max_{v \in A} C_{iv}. \tag{2}$$

The set function $F$ is *submodular*, since for all $j \in V$ and sets $A \subseteq B \subseteq V \setminus \{j\}$, we have $F(A \cup \{j\}) - F(A) \geq F(B \cup \{j\}) - F(B)$, that is, the gain of an element is diminishes as we add elements. Since the entries of $C$ are nonnegative, $F$ is *monotone*, since for all $A \subseteq B \subseteq V$, we have $F(A) \leq F(B)$. We also have $F$ *normalized*, since $F(\emptyset) = 0$.

The facility location problem is NP-Hard, so we consider approximation algorithms. Like all monotone and normalized submodular functions, the greedy algorithm guarantees a $(1 - 1/e)$-factor approximation to the optimal solution [35]. The greedy algorithm starts with the empty set, then for $k$ iterations adds the element with the largest reward. This approximation is the best possible—the maximum coverage problem is an instance of the submodular facility location problem, which was shown to be NP-Hard to optimize within a factor of $1 - 1/e + \varepsilon$ for all $\varepsilon > 0$ [13].

The problem is that the greedy algorithm has super-quadratic running time $\Theta(nmk)$ and in many datasets $n$ and $m$ can be in the millions. For this reason, several recent papers have focused on accelerating the greedy algorithm. In [26], the authors point out that if the benefit matrix is sparse, this can dramatically speed up the computation time. Unfortunately, in many problems of interest, data similarities or rewards are not sparse. Wei et al. [40] proposed to first *sparsify* the benefit matrix and then run the greedy algorithm on this new sparse matrix. In particular, [40] considers $t$-nearest neighbor sparsification, i.e., keeping for each row the $t$ largest entries and zeroing out the rest. Using this technique they demonstrated an impressive 80-fold speedup over the greedy algorithm with little loss in solution quality. One limitation of their theoretical analysis was the limited setting under which provable approximation guarantees were established.

**Our Contributions:** Inspired by the work of Wei et al. [40] we improve the theoretical analysis of the approximation error induced by sparsification. Specifically, the previous analysis assumes that the input came from a probability distribution where the preferences of each element of $i \in I$ are independently chosen uniformly at random. For this distribution, when $k = \Omega(n)$, they establish that the sparsity can be taken to be $O(\log n)$ and running the greedy algorithm on the sparsified problem will guarantee a constant factor approximation with high probability. We improve the analysis in the following ways:

- We prove guarantees for all values of $k$ and our guarantees do not require any assumptions on the input besides nonnegativity of the benefit matrix.

- In the case where $k = \Omega(n)$, we show that it is possible to take the sparsity of each row as low as $O(1)$ while guaranteeing a constant factor approximation.

- Unlike previous work, our analysis does not require the use of any particular algorithm and can be integrated to many algorithms for solving facility location problems.

- We establish a lower bound which shows that our approximation guarantees are tight up to log factors, for all desired approximation factors.

In addition to the above results we propose a novel algorithm that uses a threshold based sparsification where we keep matrix elements that are above a set value threshold. This type of sparsification is easier to efficiently implement using nearest neighbor methods. For this method of sparsification, we obtain worst case guarantees and a lower bound that matches up to constant factors. We also obtain a data dependent guarantee which helps explain why our algorithm empirically performs better than the worst case.

Further, we propose the use of Locality Sensitive Hashing (LSH) and random walk methods to accelerate approximate nearest neighbor computations. Specifically, we use two types of similarity metrics: inner products and personalized PageRank (PPR). We propose the use of fast approximations for these metrics and empirically show that they dramatically improve running times. LSH functions are well-known but, to the best of our knowledge, this is the first time they have been used to accelerate facility location problems. Furthermore, we utilize personalized PageRank as the similarity between vertices on a graph. Random walks can quickly approximate this similarity and we demonstrate that it yields highly interpretable results for real datasets.

## 2 Related Work

The use of a sparsified proxy function was shown by Wei et al. to also be useful for finding a subset for training nearest neighbor classifiers [41]. Further, they also show a connection of nearest neighbor classifiers to the facility location function. The facility location function was also used by Mirzasoleiman et al. as part of a summarization objective function in [32], where they present a summarization algorithm that is able to handle a variety of constraints.

The stochastic greedy algorithm was shown to get a $1 - 1/e - \varepsilon$ approximation with runtime $O(nm \log \frac{1}{\varepsilon})$, which has no dependance on $k$ [33]. It works by choosing a sample set from $V$ of size $\frac{n}{k} \log \frac{1}{\varepsilon}$ each iteration and adding to the current set the element of the sample set with the largest gain.

Also, there are several related algorithms for the streaming setting [5] and distributed setting [6, 25, 31, 34]. Since the objective function is defined over the entire dataset, optimizing the submodular facility location function becomes more complicated in these memory limited settings. Often the function is estimated by considering a randomly chosen subset from the set $I$.

### 2.1 Benefits Functions and Nearest Neighbor Methods

For many problems, the elements $V$ and $I$ are vectors in some feature space where the benefit matrix is defined by some similarity function sim. For example, in $\mathbb{R}^d$ we may use the RBF kernel $\text{sim}(x, y) = e^{-\gamma \|x-y\|_2^2}$, dot product $\text{sim}(x, y) = x^T y$, or cosine similarity $\text{sim}(x, y) = \frac{x^T y}{\|x\|\|y\|}$.

There has been decades of research on nearest neighbor search in geometric spaces. If the vectors are low dimensional, then classical techniques such as kd-trees [7] work well and are exact. However it has been observed that as the dimensions grow that the runtime of all known exact methods does little better than a linear scan over the dataset.

As a compromise, researchers have started to work on approximate nearest neighbor methods, one of the most successful approaches being locality sensitive hashing [15, 20]. LSH uses a hash function that hashes together items that are close. Locality sensitive hash functions exist for a variety of metrics and similarities such as Euclidean [11], cosine similarity [3, 9], and dot product [36, 38]. Nearest neighbor methods other than LSH that have been shown to work for machine learning problems include [8, 10]. Additionally, see [14] for efficient and exact GPU methods.

An alternative to vector functions is to use similarities and benefits defined from graph structures. For instance, we can use the *personalized PageRank* of vertices in a graph to define the benefit matrix [37]. The personalized PageRank is similar to the classic PageRank, except the random jumps, rather than going to anywhere in the graph, go back to the users "home" vertex. This can be used as a value of "reputation" or "influence" between vertices in a graph [17].

There are a variety of algorithms for finding the vertices with a large PageRank personalized to some vertex. One popular one is the random walk method. If $\pi_i$ is the personalized PageRank vector to some vertex $i$, then $\pi_i(v)$ is the same as the probability that a random walk of geometric length starting from $i$ ends on a vertex $v$ (where the parameter of the geometric distribution is defined by the probability of jumping back to $i$) [4]. Using this approach, we can quickly estimate all elements in the benefit matrix greater than some value $\tau$.

## 3 Guarantees for $t$-Nearest Neighbor Sparsification

We associate a bipartite support graph $G = (V, I, E)$ by having an edge between $v \in V$ and $i \in I$ whenever $C_{ij} > 0$. If the support graph is sparse, we can use the graph to calculate the gain of an element much more efficiently, since we only need to consider the neighbors of the element versus the entire set $I$. If the average degree of a vertex $i \in I$ is $t$, (and we use a cache for the current best value of an element $i$) then we can execute greedy in time $O(mtk)$. See Algorithm 1 in the Appendix for pseudocode. If the sparsity $t$ is much smaller than the size of $V$, the runtime is greatly improved.

However, the instance we wish to optimize may not be sparse. One idea is to sparsify the original matrix by only keeping the values in the benefit matrix $C$ that are $t$-nearest neighbors, which was considered in [40]. That is, for every element $i$ in $I$, we only keep the top $t$ elements of $C_{i1}, C_{i2}, \ldots, C_{in}$ and set the rest equal to zero. This leads to a matrix with $mt$ nonzero elements. We

then want the solution from optimizing the sparse problem to be close to the value of the optimal solution in the original objective function $F$.

Our main theorem is that we can set the sparsity parameter $t$ to be $O(\frac{n}{\alpha k} \log \frac{m}{\alpha k})$—which is a significant improvement for large enough $k$—while still having the solution to the sparsified problem be at most a factor of $\frac{1}{1+\alpha}$ from the value of the optimal solution.

**Theorem 1.** *Let $O_t$ be the optimal solution to an instance of the submodular facility location problem with a benefit matrix that was sparsified with $t$-nearest neighbor. For any $t \geq t^*(\alpha) = O(\frac{n}{\alpha k} \log \frac{n}{\alpha k})$, we have $F(O_t) \geq \frac{1}{1+\alpha}\text{OPT}$.*

*Proof Sketch.* For the value of $t$ chosen, there exists a set $\Gamma$ of size $\alpha k$ such that every element of $I$ has a neighbor in the $t$-nearest neighbor graph; this is proven using the probabilistic method. By appending $\Gamma$ to the optimal solution and using the monotonicity of $F$, we can move to the sparsified function, since no element of $I$ would prefer an element that was zeroed out in the sparsified matrix as one of their top $t$ most beneficial elements is present in the set $\Gamma$. The optimal solution appended with $\Gamma$ is a set of size $(1 + \alpha)k$. We then bound the amount that the optimal value of a submodular function can increase by when adding $\alpha k$ elements. See the appendix for the complete proof. □

Note that Theorem 1 is agnostic to the algorithm used to optimize the sparsified function, and so if we use a $\rho$-approximation algorithm, then we are at most a factor of $\frac{\rho}{1+\alpha}$ from the optimal solution. Later this section we will utilize this to design a subquadratic algorithm for optimizing facility location problems as long as we can quickly compute $t$-nearest neighbors and $k$ is large enough.

If $m = O(n)$ and $k = \Omega(n)$, we can achieve a constant factor approximation even when taking the sparsity parameter as low as $t = O(1)$, which means that the benefit matrix $C$ has only $O(n)$ nonzero entries. Also note that the only assumption we need is that the benefits between elements are nonnegative. When $k = \Omega(n)$, previous work was only able to take $t = O(\log n)$ and required the benefit matrix to come from a probability distribution [40].

Our guarantee has two regimes depending on the value of $\alpha$. If we want the optimal solution to the sparsified function to be a $1 - \varepsilon$ factor from the optimal solution to the original function, we have that $t^*(\varepsilon) = O(\frac{n}{\varepsilon k} \log \frac{m}{\varepsilon k})$ suffices. Conversely, if we want to take the sparsity $t$ to be much smaller than $\frac{n}{k} \log \frac{m}{k}$, then this is equivalent to taking $\alpha$ very large and we have some guarantee of optimality.

In the proof of Theorem 1, the only time we utilize the value of $t$ is to show that there exists a small set $\Gamma$ that covers the entire set $I$ in the $t$-nearest neighbor graph. Real datasets often contain a covering set of size $\alpha k$ for $t$ much smaller than $O(\frac{n}{\alpha k} \log \frac{m}{\alpha k})$. This observation yields the following corollary.

**Corollary 2.** *If after sparsifying a problem instance there exists a covering set of size $\alpha k$ in the $t$-nearest neighbor graph, then the optimal solution $O_t$ of the sparsified problem satisfies $F(O_t) \geq \frac{1}{1+\alpha}\text{OPT}$.*

In the datasets we consider in our experiments of roughly 7000 items, we have covering sets with only 25 elements for $t = 75$, and a covering set of size 10 for $t = 150$. The size of covering set was upper bounded by using the greedy set cover algorithm. In Figure 2 in the appendix, we see how the size of the covering set changes with the choice of the number of neighbors chosen $t$.

It would be desirable to take the sparsity parameter $t$ lower than the value dictated by $t^*(\alpha)$. As demonstrated by the following lower bound, is not possible to take the sparsity significantly lower than $\frac{1}{\alpha}\frac{n}{k}$ and still have a $\frac{1}{1+\alpha}$ approximation in the worst case.

**Proposition 3.** *Suppose we take*

$$ t = \max\left\{\frac{1}{2\alpha}, \frac{1}{1+\alpha}\right\} \frac{n-1}{k}. $$

*There exists a family of inputs such that we have $F(O_t) \leq \frac{1}{1+\alpha-2/k}\text{OPT}$.*

The example we create to show this has the property that in the $t$-nearest neighbor graph, the set $\Gamma$ needs $\alpha k$ elements to cover every element of $I$. We plant a much smaller covering set that is very close in value to $\Gamma$ but is hidden after sparsification. We then embed a modular function within the facility location objective. With knowledge of the small covering set, an optimal solver can take

advantage of this modular function, while the sparsified solution would prefer to first choose the set $\Gamma$ before considering the modular function. See the appendix for full details.

Sparsification integrates well with the stochastic greedy algorithm [33]. By taking $t \geq t^*(\varepsilon/2)$ and running stochastic greedy with sample sets of size $\frac{n}{k} \ln \frac{2}{\varepsilon}$, we get a $1 - 1/e - \varepsilon$ approximation in expectation that runs in expected time $O(\frac{nm}{\varepsilon k} \log \frac{1}{\varepsilon} \log \frac{m}{\varepsilon k})$. If we can quickly sparsify the problem and $k$ is large enough, for example $n^{1/3}$, this is subquadratic. The following proposition shows a high probability guarantee on the runtime of this algorithm and is proven in the appendix.

**Proposition 4.** *When* $m = O(n)$, *the stochastic greedy algorithm [33] with set sizes of size* $\frac{n}{k} \log \frac{2}{\varepsilon}$, *combined with sparsification with sparsity parameter* $t$, *will terminate in time* $O(n \log \frac{1}{\varepsilon} \max\{t, \log n\})$ *with high probability. When* $t \geq t^*(\varepsilon/2) = O(\frac{n}{\varepsilon k} \log \frac{m}{\varepsilon k})$, *this algorithm has a* $1 - 1/e - \varepsilon$ *approximation in expectation.*

## 4 Guarantees for threshold-based Sparsification

Rather than $t$-nearest neighbor sparsification, we now consider using $\tau$-threshold sparsification, where we zero-out all entries that have value below a threshold $\tau$. Recall the definition of a locality sensitive hash.

**Definition.** $\mathcal{H}$ is a $(\tau, K\tau, p, q)$-locality sensitive hash family if for $x, y$ satisfying $\text{sim}(x, y) \geq \tau$ we have $\text{P}_{h \in \mathcal{H}}(h(x) = h(y)) \geq p$ and if $x, y$ satisfy $\text{sim}(x, y) \leq K\tau$ we have $\text{P}_{h \in \mathcal{H}}(h(x) = h(y)) \leq q$.

We see that $\tau$-threshold sparsification is a better model than $t$-nearest neighbors for LSH, as for $K = 1$ it is a noisy $\tau$-sparsification and for non-adversarial datasets it is a reasonable approximation of a $\tau$-sparsification method. Note that due to the approximation constant $K$, we do not have an *a priori* guarantee on the runtime of arbitrary datasets. However we would expect in practice that we would only see a few elements with threshold above the value $\tau$. See [2] for a discussion on this.

One issue is that we do not know how to choose the threshold $\tau$. We can sample elements of the benefit matrix $C$ to estimate how sparse the threshold graph will be for a given threshold $\tau$. Assuming the values of $C$ are in general position[1], by using the Dvoretzky-Kiefer-Wolfowitz-Massart Inequality [12, 28] we can bound the number of samples needed to choose a threshold that achieves a desired sparsification level.

We establish the following data-dependent bound on the difference in the optimal solutions of the $\tau$-threshold sparsified function and the original function. We denote the set of vertices adjacent to $S$ in the $\tau$-threshold graph with $N(S)$.

**Theorem 5.** *Let* $O_\tau$ *be the optimal solution to an instance of the facility location problem with a benefit matrix that was sparsified using a* $\tau$-threshold. *Assume there exists a set* $S$ *of size* $k$ *such that in the* $\tau$-threshold graph we have the neighborhood of $S$ satisfying $|N(S)| \geq \mu n$. *Then we have*

$$F(O_\tau) \geq \left(1 + \frac{1}{\mu}\right)^{-1} \text{OPT}.$$

For the datasets we consider, we see that we can keep a $0.01 - 0.001$ fraction of the elements of $C$ while still having a small set $S$ with a neighborhood $N(S)$ that satisfied $|N(S)| \geq 0.3n$. In Figure 3 in the appendix, we plot the relationship between the number of edges in the $\tau$-threshold graph and the number of coverable element by a a set of small size, as estimated by the greedy algorithm for max-cover.

Additionally, we have worst case dependency on the number of edges in the $\tau$-threshold graph and the approximation factor. The guarantees follow from applying Theorem 5 with the following Lemma.

**Lemma 6.** *For* $k \geq \frac{c}{1-2c^2} \frac{1}{\delta}$, *any graph with* $\frac{1}{2} \delta^2 n^2$ *edges has a set* $S$ *of size* $k$ *such that the neighborhood* $N(S)$ *satisfies*

$$|N(S)| \geq c\delta n.$$

To get a matching lower bound, consider the case where the graph has two disjoint cliques, one of size $\delta n$ and one of size $(1 - \delta)n$. Details are in the appendix.

## 5   Experiments

### 5.1   Summarizing Movies and Music from Ratings Data

We consider the problem of summarizing a large collection of movies. We first need to create a feature vector for each movie. Movies can be categorized by the people who like them, and so we create our feature vectors from the MovieLens ratings data [16]. The MovieLens database has 20 million ratings for 27,000 movies from 138,000 users. To do this, we perform low-rank matrix completion and factorization on the ratings matrix [21, 22] to get a matrix $X = UV^T$, where $X$ is the completed ratings matrix, $U$ is a matrix of feature vectors for each user and $V$ is a matrix of feature vectors for each movie. For movies $i$ and $j$ with vectors $v_i$ and $v_j$, we set the benefit function $C_{ij} = v_i^T v_j$. We do not use the normalized dot product (cosine similarity) because we want our summary movies to be movies that were highly rated, and not normalizing makes highly rated movies increase the objective function more.

We complete the ratings matrix using the MLlib library in Apache Spark [29] after removing all but the top seven thousand most rated movies to remove noise from the data. We use locality sensitive hashing to perform sparsification; in particular we use the LSH in the FALCONN library for cosine similarity [3] and the reduction from a cosine simiarlity hash to a dot product hash [36]. As a baseline we consider sparsification using a scan over the entire dataset, the stochastic greedy algorithm with lazy evaluations[33], and the greedy algorithm with lazy evaluations [30]. The number of elements chosen was set to 40 and for the LSH method and stochastic greedy we average over five trials.

We then do a scan over the sparsity parameter $t$ for the sparsification methods and a scan over the number of samples drawn each iteration for the stochastic greedy algorithm. The sparsified methods use the (non-stochastic) lazy greedy algorithm as the base optimization algorithm, which we found worked best for this particular problem[2]. In Figure 1(a) we see that the LSH method very quickly approaches the greedy solution—it is almost identical in value just after a few seconds even though the value of $t$ is much less than $t^*(\varepsilon)$. The stochastic greedy method requires much more time to get the same function value. Lazy greedy is not plotted, since it took over 500 seconds to finish.

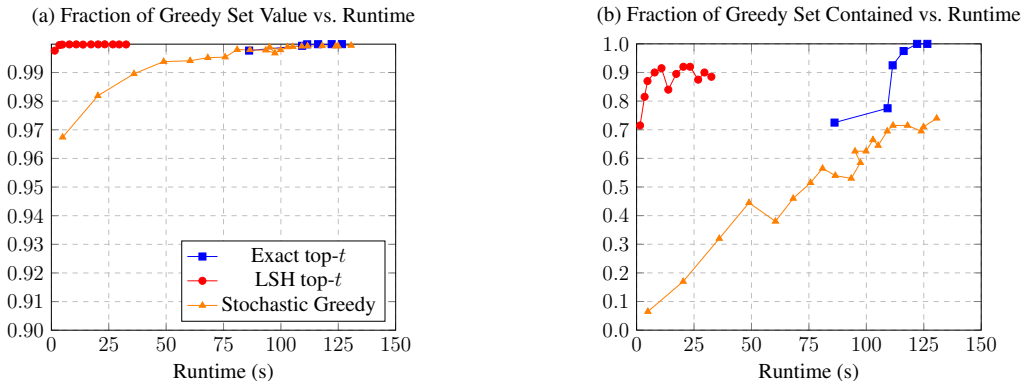

Figure 1: Results for the MovieLens dataset [16]. Figure (a) shows the function value as the runtime increases, normalized by the value the greedy algorithm obtained. As can be seen our algorithm is within 99.9% of greedy in less than 5 seconds. For this experiment, the greedy algorithm had a runtime of 512 seconds, so this is a 100x speed up for a small penalty in performance. We also compare to the stochastic greedy algorithm [33], which needs 125 seconds to get equivalent performance, which is 25x slower.
Figure (b) shows the fraction of the set that was returned by each method that was common with the set returned by greedy. We see that the approximate nearest neighbor method has 90% of its elements common with the greedy set while being 50x faster than greedy, and using exact nearest neighbors can perfectly match the greedy set while being 4x faster than greedy.

Table 1: A subset of the summarization outputted by our algorithm on the MovieLens dataset, plus the elements who are represented by each representative with the largest dot product. Each group has a natural interpretation: 90's slapstick comedies, 80's horror, cult classics, etc. Note that this was obtained with only a similarity matrix obtained from ratings.

| Happy Gilmore | Nightmare on Elm Street | Star Wars IV | Shawshank Redemption |
|---|---|---|---|
| Tommy Boy | Friday the 13th | Star Wars V | Schindler's List |
| Billy Madison | Halloween II | Raiders of the Lost Ark | The Usual Suspects |
| Dumb & Dumber | Nightmare on Elm Street 3 | Star Wars VI | Life Is Beautiful |
| Ace Ventura Pet Detective | Child's Play | Indiana Jones, Last Crusade | Saving Private Ryan |
| Road Trip | Return of the Living Dead II | Terminator 2 | American History X |
| American Pie 2 | Friday the 13th 2 | The Terminator | The Dark Knight |
| Black Sheep | Puppet Master | Star Trek II | Good Will Hunting |

| Pulp Fiction | The Notebook | Pride and Prejudice | The Godfather |
|---|---|---|---|
| Reservoir Dogs | P.S. I Love You | Anne of Green Gables | The Godfather II |
| American Beauty | The Holiday | Persuasion | One Flew Over the Cuckoo's Nest |
| A Clockwork Orange | Remember Me | Emma | Goodfellas |
| Trainspotting | A Walk to Remembe | Mostly Martha | Apocalypse Now |
| Memento | The Proposal | Desk Set | Chinatown |
| Old Boy | The Vow | The Young Victoria | 12 Angry Men |
| No Country for Old Men | Life as We Know It | Mansfield Park | Taxi Driver |

A performance metric that can be better than the objective value is the fraction of elements returned that are common with the greedy algorithm. We treat this as a proxy for the interpretability of the results. We believe this metric is reasonable since we found the subset returned by the greedy algorithm to be quite interpretable. We plot this metric against runtime in Figure 1b. We see that the LSH method quickly gets to 90% of the elements in the greedy set while stochastic greedy takes much longer to get to just 70% of the elements. The exact sparsification method is able to completely match the greedy solution at this point. One interesting feature is that the LSH method does not go much higher than 90%. This may be due to the increased inaccuracy when looking at elements with smaller dot products. We plot this metric against the number of exact and approximate nearest neighbors $t$ in Figure 4 in the appendix.

We include a subset of the summarization and for each representative a few elements who are represented by this representative with the largest dot product in Table 1 to show the interpretability of our results.

## 5.2 Finding Influential Actors and Actresses

For our second experiment, we consider how to find a diverse subset of actors and actresses in a collaboration network. We have an edge between an actor or actress if they collaborated in a movie together, weighted by the number of collaborations. Data was obtained from [19] and an actor or actress was only included if he or she was one of the top six in the cast billing. As a measure of influence, we use personalized PageRank [37]. To quickly calculate the people with the largest influence relative to someone, we used the random walk method[4].

We first consider a small instance where we can see how well the sparsified approach works. We build a graph based on the cast in the top thousand most rated movies. This graph has roughly 6000 vertices and 60,000 edges. We then calculate the entire PPR matrix using the power method. Note that this is infeasible on larger graphs in terms of time and memory. Even on this moderate sized graph it took six hours and takes up two gigabytes of space. We then compare the value of the greedy algorithm using the entire PPR matrix with the sparsified algorithm using the matrix approximated by Monte Carlo sampling using the two metrics mentioned in the previous section. We omit exact nearest neighbor and stochastic greedy because it is not clear how it would work without having to compute the entire PPR matrix. Instead we compare to an approach where we choose a sample from $I$ and calculate the PPR only on these elements using the power method. As mentioned in Section 2, several algorithms utilize random sampling from $I$. We take $k$ to be 50 for this instance. In Figure 5 in the appendix we see that sparsification performs drastically better in both function value and percent of the greedy set contained for a given runtime.

We now scale up to a larger graph by taking the actors and actresses billed in the top six for the twenty thousand most rated movies. This graph has 57,000 vertices and 400,000 edges. We would not be able to compute the entire PPR matrix for this graph in a reasonable amount of time and even if we could it would be a challenge to store the entire matrix in memory. However we can run the sparsified algorithm in three hours using only 2 GB of memory, which could be improved further by parallelizing the Monte Carlo approximation.

We run the greedy algorithm separately on the actors and actresses. For each we take the top twenty-five and compare to the actors and actresses with the largest (non-personalized) PageRank. In Figure 2 of the appendix, we see that the PageRank output fails to capture the diversity in nationality of the dataset, while the submodular facility location optimization returns actors and actresses from many of the worlds film industries.

## Acknowledgements

This material is based upon work supported by the National Science Foundation Graduate Research Fellowship under Grant No. DGE-1110007 as well as NSF Grants CCF 1344179, 1344364, 1407278, 1422549 and ARO YIP W911NF-14-1-0258.

## Footnotes

[1]By this we mean that the values of $C$ are all unique, or at least only a few elements take any particular value. We need this to hold since otherwise a threshold based sparsification may exclusively return an empty graph or the complete graph.

[2]When experimenting on very larger datasets, we found that runtime constraints can make it necessary to use stochastic greedy as the base optimization algorithm

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
