[Supplementary Material]

# 6 Appendix: Additional Figures

---

**Algorithm 1** Greedy algorithm with sparsity graph

---
Input: benefit matrix $C$, sparsity graph $G = (V, I, E)$
define $N(v)$: return the neighbors of $v$ in $G$
for all $i \in I$:
    # cache of the current benefit given to $i$
    $\beta_i \leftarrow 0$
$A \leftarrow \emptyset$
for $k$ iterations:
    for all $v \in V$:
        # calculate the gain of element $v$
        $g_v \leftarrow 0$
        for all $i \in N(v)$:
            # add the gain of element $v$ from $i$
            $g_v \leftarrow g_v + \max(C_{iv} - \beta_i, 0)$
    $v^* \leftarrow \arg\max_V g_v$
    $A \leftarrow A \cup \{v^*\}$
    for all $i \in N(v^*)$
        # update the cache of the current benefit for $i$
        $\beta_i \leftarrow \max(\beta_i, C_{iv^*})$
return $A$

---

Figure 2: (a) MovieLens Dataset [16] and (b) IMDb Dataset [19] as explained in the Experiments Section. We see that for sparsity $t$ significantly smaller than the $n/(\alpha k)$ lower bound we can still find a small covering set in the $t$-nearest neighbor graph.

(a) MovieLens

(b) IMDb

Figure 3: (a) MovieLens Dataset [16] and (b) IMDb Dataset [19], as explained in the Experiments Section. We see that even with several orders of magnitude fewer edges than the complete graph we still can find a small set that covers a large fraction of the dataset. For MovieLens this set was of size 40 and for IMDb this set was of size 50. The number of coverable was estimated by the greedy algorithm for the max-coverage problem.

Figure 4: The fraction of the greedy solution that was contained as the value of the sparsity $t$ was increased for exact nearest neighbor and approximate LSH-based nearest neighbor on the MovieLens dataset. We see that the exact method captures slightly more of greedy solution for a given value of $t$ and the LSH value does not converge to 1. However LSH still captures a reasonable amount of the greedy set and is significantly faster at finding nearest neighbors.

(a) Fraction of Greedy Set Value vs. Runtime

(b) Fraction of Greedy Set Contained vs. Runtime

Figure 5: Results for the IMDb dataset [19]. Figure (a) shows the function value as the runtime increases, normalized by the value the greedy algorithm obtained. As can be seen our algorithm is within 99% of greedy in less than 10 minutes. For this experiment, the greedy algorithm had a runtime of six hours, so this is a 36x acceleration for a small penalty in performance. We also compare to using a small sample of the set $I$ as an estimate of the function, which does not perform nearly as well as our algorithm even for much longer time.

Figure (b) shows the fraction of the set that was returned by each method that was common with the set returned by greedy. We see that the approximate nearest neighbor method has 90% of its elements common with the greedy set while being 18x faster than greedy.

Table 2: The top twenty-five actors and actresses generated by sparsified facility location optimization defined by the personalized PageRank of a 57,000 vertex movie personnel collaboration graph from [19] and the twenty-five actors and actresses with the largest (non-personalized) PageRank. We see that the classical PageRank approach fails to capture the diversity of nationality in the dataset, while the facility location results have actors and actresses from many of the worlds film industries.

| Actors | | Actresses | |
| --- | --- | --- | --- |
| Facility Location | PageRank | Facility Location | PageRank |
| Robert De Niro | Jackie Chan | Julianne Moore | Julianne Moore |
| Jackie Chan | Gérard Depardieu | Susan Sarandon | Susan Sarandon |
| Gérard Depardieu | Robert De Niro | Bette Davis | Juliette Binoche |
| Kemal Sunal | Michael Caine | Isabelle Huppert | Isabelle Huppert |
| Shah Rukh Khan | Samuel L. Jackson | Kareena Kapoor | Catherine Deneuve |
| Michael Caine | Christopher Lee | Juliette Binoche | Kristin Scott Thomas |
| John Wayne | Donald Sutherland | Meryl Streep | Meryl Streep |
| Samuel L. Jackson | Peter Cushing | Adile Naşit | Bette Davis |
| Bud Spencer | Nicolas Cage | Catherine Deneuve | Nicole Kidman |
| Peter Cushing | John Wayne | Li Gong | Charlotte Rampling |
| Toshirô Mifune | John Cusack | Helena Bonham Carter | Helena Bonham Carter |
| Steven Seagal | Christopher Walken | Penélope Cruz | Kathy Bates |
| Moritz Bleibtreu | Bruce Willis | Naomi Watts | Naomi Watts |
| Jean-Claude Van Damme | Kemal Sunal | Masako Nozawa | Cate Blanchett |
| Mads Mikkelsen | Harvey Keitel | Drew Barrymore | Drew Barrymore |
| Michael Ironside | Amitabh Bachchan | Charlotte Rampling | Helen Mirren |
| Amitabh Bachchan | Shah Rukh Khan | Golshifteh Farahani | Michelle Pfeiffer |
| Ricardo Darín | Sean Bean | Hanna Schygulla | Penélope Cruz |
| Charles Chaplin | Steven Seagal | Toni Collette | Sigourney Weaver |
| Sean Bean | Jean-Claude Van Damme | Kati Outinen | Toni Collette |
| Louis de Funès | Morgan Freeman | Edna Purviance | Catherine Keener |
| Tadanobu Asano | Christian Slater | Monica Bellucci | Heather Graham |
| Bogdan Diklic | Val Kilmer | Kristin Scott Thomas | Sandra Bullock |
| Nassar | Liam Neeson | Catherine Keener | Kirsten Dunst |
| Lance Henriksen | Gene Hackman | Kyôko Kagawa | Miranda Richardson |

# 7 Appendix: Full Proofs

## 7.1 Proof of Theorem 1

We will use the following two lemmas in the proof of Theorem 1, which are proven later in this section. The first lemma bounds the size of the smallest set of left vertices covering every right vertex in a $t$-regular bipartite graph.

**Lemma 7.** *For any bipartite graph $G = (V, I, E)$ such that $|V| = n$, $|I| = m$, every vertex $i \in I$ has degree at least $t$, and $n \leq mt$, there exists a set of vertices $\Gamma \subseteq V$ such that every vertex in $I$ has a neighbor in $\Gamma$ and*

$$|\Gamma| \leq \frac{n}{t}\left(1 + \ln\frac{mt}{n}\right). \tag{3}$$

The second lemma bounds the rate that the optimal solution grows as a function of $k$.

**Lemma 8.** *Let $f$ be any normalized submodular function and let $O_2$ and $O_1$ be optimal solutions for their respective sizes, with $|O_2| \geq |O_1|$. We have*

$$f(O_2) \leq \frac{|O_2|}{|O_1|}f(O_1).$$

We now prove Theorem 1.

*Proof.* We will take $t^*(\alpha)$ to be the smallest value of $t$ such that $|\Gamma| \geq \alpha k$ in Equation (3). It can be verified that $t^*(\alpha) \leq \lceil 4\frac{n}{\alpha k}\max\{1, \ln\frac{n}{\alpha k}\}\rceil$.

Let $\Gamma \subseteq V$ be a set such that all elements of $I$ has a $t$-nearest neighbor in $\Gamma$. By Lemma 7, one is guaranteed to exists of size at most $\alpha k$ for $t \geq t^*(\alpha)$. Let $O$ be the optimal set of size $k$ and let $F^{(t)}$ be the objective function of the sparsified function. Let $O_t^k$ and $O_t^{(1+\alpha)k}$ be the optimal solutions to $F^{(t)}$ of size $k$ and $(1+\alpha)k$. We have

$$
\begin{aligned}
F(O) &\leq F(O \cup \Gamma) \\
&= F^{(t)}(O \cup \Gamma) \\
&\leq F^{(t)}(O_t^{(1+\alpha)k}).
\end{aligned}
\tag{4}
$$

The first inequality is due to the monotonicity of $F$. The second is because every element of $I$ would prefer to choose one of their $t$ nearest neighbors and because of $\Gamma$ they can. The third inequality is because $|O \cup \Gamma| \leq (1+\alpha)k$ and $O_t^{(1+\alpha)k}$ is the optimal solution for this size.

Now by Lemma 8, we can bound the approximation for shrinking from $O_t^{(1+\alpha)k}$ to $O_t^k$. Applying Lemma 8 and continuing from Equation (4) implies

$$
F(O) \leq (1+\alpha)F^{(t)}(O_t^k).
$$

Observe that $F^{(t)}(A) \leq F(A)$ for any set $A$ to obtain the final bound. $\square$

## 7.2 Proof of Proposition 3

Define $\Pi_n(t)$ to be the $n \times (n+1)$ matrix where for $i = 1, \ldots, n$ we have column $i$ equal to 1 for positions $i$ to $i + t - 1$, potentially cycling the position back to the beginning if necessary, and then 0 otherwise. For column $n + 1$ make all values $1 - 1/2n$. For example,

$$
\Pi_6(3) = \begin{pmatrix}
1 & 0 & 0 & 0 & 1 & 1 & 11/12 \\
1 & 1 & 0 & 0 & 0 & 1 & 11/12 \\
1 & 1 & 1 & 0 & 0 & 0 & 11/12 \\
0 & 1 & 1 & 1 & 0 & 0 & 11/12 \\
0 & 0 & 1 & 1 & 1 & 0 & 11/12 \\
0 & 0 & 0 & 1 & 1 & 1 & 11/12
\end{pmatrix}.
$$

We will show the lower bound in two parts, when $\alpha < 1$ and when $\alpha \geq 1$.

*Proof of case $\alpha \geq 1$.* Let $F$ be the facility location function defined on the benefit matrix $C = \Pi_n(\delta \frac{n}{k})$. For $t = \delta \frac{n}{k}$, the sparsified matrix $C^{(t)}$ has all of its elements except the $n + 1$st row. With $k$ elements, the optimal solution to $F^{(t)}$ is to choose the $k$ elements that let us cover $\delta n$ of the elements of $I$, giving a value of $\delta n$. However if we chose the $n + 1$th element, we would have gotten a value of $n - 1/2$, giving an approximation of $\frac{\delta}{1 - 1/(2n)}$. Setting $\delta = 1/(1+\alpha)$ and using $\alpha \leq n/k$ implies

$$
F(O_t) \leq \frac{1}{1 + \alpha - 1/k} \text{OPT}
$$

when we take $t = \frac{1}{1+\alpha} \frac{|V|-1}{k}$ (note that for this problem $|V| = n + 1$). $\square$

*Proof of case $\alpha < 1$.* Let $F$ be the facility location function defined on the benefit matrix

$$
C = \begin{pmatrix}
\Pi_n(\frac{1}{\alpha} \frac{n}{k}) & 0 \\
0 & (\frac{1}{\alpha} \frac{n}{k} - \frac{1}{2n}) I_{n \times n}
\end{pmatrix}
$$

For $t = \frac{1}{\alpha} \frac{n}{k}$, the optimal solution to $F^{(t)}$ is to use $\alpha k$ elements to cover all the elements of $\Pi_n$, then use the remaining $(1-\alpha)k$ elements in the identity section of the matrix. This has a value of less than $\frac{1}{\alpha} n$. For $F$, the optimal solution is to choose the $n + 1$st element of $\Pi_n$, then use the remaining $k - 1$ elements in the identity section of the identity section of the matrix. This has a value of more than $n(1 + \frac{1}{\alpha} - \frac{1}{k\alpha} - \frac{1}{n})$, and therefore an approximation of $\frac{1}{1+\alpha-1/k-1/n}$. Note that in this case $|V| = 2n + 1$ and so we have

$$
F(O_t) \leq \frac{1}{1 + \alpha - 1/k - 1/n} \text{OPT}
$$

when we take $t = \frac{1}{2\alpha} \frac{|V|-1/2}{k}$. $\square$

## 7.3  Proof of Proposition 4

*Proof.* The stochastic greedy algorithm works by choosing a set of elements $S_j$ each iteration of size $\frac{n}{k} \log \frac{1}{\varepsilon}$. We will assume $m = n$ and $\varepsilon = 1/e$ to simplify notation. We want to show that

$$\sum_{j=1}^{k} \sum_{v \in S_j} d_v = O(nt)$$

with high probability, where $d_v$ is the degree of element $v$ in the sparsity graph. We will show this using Bernstein's Inequality: given $n$ i.i.d. random variables $X_1, \ldots, X_n$ such that $\mathrm{E}(X_\ell) = 0$, $\mathrm{Var}(X_\ell) = \sigma^2$, and $|X_\ell| \leq c$ with probability 1, we have

$$\mathrm{P}\left(\sum_{\ell=1}^{n} X_\ell \geq \lambda n\right) \leq \exp\left(-\frac{n\lambda^2}{2\sigma^2 + \frac{2}{3}c\lambda}\right).$$

We will take $X_\ell$ to be the degree of the $\ell$th element of $V$ chosen uniformly at random, shifted by the mean of $t$. Although in the stochastic greedy algorithm the elements are not chosen i.i.d. but instead iterations in $k$ iterations of sampling without replacement, treating them as i.i.d. random variables for purposes of Bernstein's Inequality is justified by Theorem 4 of [18].

We have $|X_\ell| \leq n$, and $\mathrm{Var}(X_\ell) \leq tn$, where the variance bound is because variance for a given mean $t$ on support $[0, m]$ is maximized by putting mass $\frac{t}{n}$ on $n$ and $1 - \frac{t}{n}$ on 0.

If $t \geq \ln n$, then take $\lambda = \frac{8}{3}t$. If $t < \ln n$, take $\lambda = \frac{8}{3}\ln n$. This yields

$$\mathrm{P}\left(\sum_{j=1}^{k} \sum_{v \in S_j} d_v \geq nt + \frac{8}{3}\sqrt{\frac{m}{n}} \max\{nt, \ln n\}\right) \leq \frac{1}{n}.$$

$\square$

## 7.4  Proof of Lemma 7

We now prove Lemma 7, which is a modification of Theorem 1.2.2 of [1].

*Proof.* Choose a set $X$ by picking each element of $V$ with probability $p$, where $p$ is to be decided later. For every element of $I$ without a neighbor in $X$, add one arbitrarily. Call this set $Y$. We have $\mathrm{E}(|X \cup Y|) \leq np + m(1-p)^t \leq np + me^{-pt}$. Optimizing for $p$ yields $p = \frac{1}{t}\ln\frac{mt}{n}$. This is a valid probability when $\frac{mt}{n} \geq 1$, which we assumed, and when $\frac{m}{n} \leq \frac{e^t}{t}$ (we do not need to worry about the latter case because if it does not hold then it implies an inequality weaker than the trivial one $|\Gamma| \leq n$). $\square$

## 7.5  Proof of Lemma 8

Before we prove Lemma 8, we need the following Lemma.

**Lemma 9.** *Let $f$ be any normalized submodular function and let $O$ be an optimal solution for its respective size. Let $A$ be any set. We have*

$$f(O \cup A) \leq \left(1 + \frac{|A|}{|O|}\right) f(O).$$

We now prove Lemma 8.

*Proof.* Let

$$A^* = \underset{\{A \subseteq O_2 : |A| \leq |O_1|\}}{\arg\max} f(A)$$

and let $A' = O_2 \setminus A^*$. Since $A^*$ is optimal for the function when restricted to a ground set $O_2$, by Lemma 9 and the optimality of $O_1$ for sets of size $|O_1|$, we have

$$
\begin{aligned}
f(O_2) &= f(A^* \cup A') \\
&\leq \left(1 + \frac{|A'|}{|A^*|}\right) f(A^*) \\
&= \frac{|O_2|}{|O_1|} f(A^*) \\
&\leq \frac{|O_2|}{|O_1|} f(O_1).
\end{aligned}
$$

$\square$

We now prove Lemma 9.

*Proof.* Define $f(v \mid A) = f(\{v\} \cup A) - f(A)$. Let $O = \{o_1, \ldots o_k\}$, where the ordering is arbitrary except that

$$
f(o_k \mid O \setminus \{o_k\}) = \arg\min_{i=1,\ldots,k} f(o_i \mid O \setminus \{o_i\}).
$$

Let $A = \{a_1, \ldots, a_\ell\}$, where the ordering is arbitrary except that

$$
f(a_1 \mid O) = \arg\max_{i=1,\ldots,\ell} f(a_i \mid O).
$$

We will first show that

$$
f(a_1 \mid O) \leq f(o_k \mid O \setminus \{o_k\}). \tag{5}
$$

By submodularity, we have

$$
f(a_1 \mid O) \leq f(a_1 \mid O \setminus \{o_k\}).
$$

If it was true that

$$
f(a_1 \mid O \setminus \{o_k\}) > f(o_k \mid O \setminus \{o_k\}),
$$

then we would have

$$
\begin{aligned}
f((O \setminus \{o_k\}) \cup \{a_1\}) &= f(a_1 \mid O \setminus \{o_k\}) + \sum_{i=1}^{k-1} f(o_i \mid \{o_1, \ldots, o_{i-1}\}) \\
&\geq \sum_{i=1}^{k} f(o_i \mid \{o_1, \ldots, o_{i-1}\}) \\
&= f(O),
\end{aligned}
$$

contradicting the optimality of $O$, thus showing that Inequality 5 holds.

Now since for all $i \in \{1, 2, \ldots, k\}$

$$
\begin{aligned}
f(a_1 \mid O) &\leq f(o_k \mid O \setminus \{o_k\}) \\
&\leq f(o_i \mid O \setminus \{o_i\}) \\
&\leq f(o_i \mid \{o_1, \ldots, o_{i-1}\}),
\end{aligned}
$$

it is worse than the average of $f(o_i \mid \{o_1, \ldots, o_{i-1}\})$, which is $\frac{1}{k} \sum_{i=1}^{k} f(o_i \mid \{o_1, \ldots, o_{i-1}\})$, and showing that

$$
f(a_1 \mid O) \leq \frac{1}{k} f(O). \tag{6}
$$

Finally, we have

$$
\begin{aligned}
f(O \cup A) = f(O) + \sum_{i=1}^{\ell} f(a_i \mid O \cup \{a_1, \ldots, a_{i-1}\}) \\
\leq f(O) + \sum_{i=1}^{\ell} f(a_i \mid O) \\
\leq f(O) + \ell f(a_1 \mid O) \\
\leq \left(1 + \frac{\ell}{k}\right) f(O),
\end{aligned}
$$

which is what we wanted to show. □

## 7.6 Proof of Theorem 5

*Proof.* Let $O$ be the optimal solution to the original problem. Let $F_\tau$ and $\bar{F}_\tau$ be the functions defined restricting to the matrix elements with benefit at least $\tau$ and all remaining elements, respectively. If there exists a set $S$ of size $k$ such that $\mu n$ elements have a neighbor in $S$, then we have

$$
\begin{aligned}
F(O) \leq F_\tau(O) + \bar{F}_\tau(O) \\
\leq F_\tau(O) + n\tau \\
\leq F_\tau(O) + \frac{1}{\mu} F_\tau(S) \\
\leq \left(1 + \frac{1}{\mu}\right) F_\tau(O_\tau)
\end{aligned}
$$

where the last inequality follows from $O_\tau$ being optimal for $F_\tau$. □

## 7.7 Proof of Lemma 6

*Proof.* Consider the following algorithm:
$\quad B \leftarrow \emptyset$
$\quad S \leftarrow \emptyset$
$\quad \text{while } |B| \leq c\delta n$
$\quad\quad v^* \leftarrow \arg\max |N(v)|$
$\quad\quad \text{add } v^* \text{ to } S$
$\quad\quad \text{add } N(v^*) \text{ to } B$
$\quad\quad \text{remove } N(v^*) \cup \{v^*\} \text{ from } G$

We will show that after $T = \frac{c}{(1-2c^2)\delta}$ iterations this algorithm will terminate. When it does, $S$ will satisfy $|N(S)| \geq c\delta n$ since every element of $B$ has a neighbor in $S$.

If there exists a vertex of degree $c\delta n$, then we will terminate after the first iteration. Otherwise all vertices have degree less than $c\delta n$. Assuming all vertices have degree less than $c\delta n$, until we terminate the number of edges incident to $B$ is at most $|B|c\delta n \leq c^2\delta^2 n^2$. At each iteration, the number of edges in the graph is at least $(\frac{1}{2} - c^2)\delta^2 n^2$, thus in each iteration we can find a $v^*$ with degree at least $(1 - 2c^2)\delta^2 n$. Therefore, after $T$ iterations, we will have terminated with the size of $S$ is at most $T$ and $|N(S)| \geq c\delta n$. □

We see that this is tight up to constant factors by the following proposition.

**Proposition 10.** *There exists an example where for $\Delta = \delta^2 n$, the optimal solution to the sparsified function is a factor of $O(\delta)$ from the optimal solution to the original function.*

*Proof.* Consider the following benefit matrix.

$$
C = \begin{pmatrix} \mathbb{1}_{\delta n \times \delta n} + (1 + \frac{1}{k-1})I & 0 \\ 0 & (1 - \frac{1}{(1-\delta)n})\mathbb{1}_{(1-\delta n) \times (1-\delta n)} \end{pmatrix}
$$

The sparsified optimal would only choose elements in the top left clique and would get a value of roughly $\delta n$, while the true optimal solution would cover both cliques and get a value of roughly $n$. $\qquad\square$