[Reviews · NeurIPS 2016]

Reviewer 1

Summary

This paper studies the sparsifying the facility location problems. Motivated by Wei et al 2014, they show that the facility location similarity graph may be significantly sparsified without incurring a significant loss on its optimization performance. In the case of t-nearest neighbor graph, the authors demonstrate that constant factor approximation is possible even if the similarity graph is as sparse as t=O(1) given the condition that k (size of the desired summary) is in the order of n. They also demonstrate the lower bound matching up to a log factor. In the case of \tau threshold sparsification, the authors give a data dependent bound which is also matched up to a constant. They show that \tau threshold sparsification is useful in combination with LSH. They tested both sparsification methods (t-nearest neighbor and \tau thresholding) in the MovieLens and IMDb data sets. Significant speedup was shown by both sparsification methods and interesting interpretable results on these data sets are shown in the resulting summaries.

Qualitative Assessment

This paper studies an important problem -- scaling up optimizing the facility location function, which has a number of data summarization applications. The bounds for both the t-nearest neighbor case and the \tau-thresholding case are neat, interesting, and tight. I think the paper may be of interests to many people working in this area and should be accepted. A few suggestions and questions: - It would be better to clarify the range of \alpha in line 129 and line 197. Based on my understanding, \alpha > 0 is implied for line 129, and 0 < \alpha < 1 is implied for line 197. - In line 161, it is worth mentioning how to find the smallest covering set for a given sparsity t. - It is unclear to me which greedy algorithm is used in the experiments. Is it the lazy greedy or the naive greedy? - I think a better experimental setup is to fix the optimization algorithm (either naive greedy, lazy greedy or stochastic greedy), and then compare between full similarity compute and the two different sparsification methods - A relevant paper to the facility location function below that the authors should cite: Wei, Kai, Rishabh Iyer, and Jeff Bilmes. "Submodularity in data subset selection and active learning." Proceedings of the 32nd International Conference on Machine Learning, Lille, Fran. 2015.

Confidence in this Review

3-Expert (read the paper in detail, know the area, quite certain of my opinion)


Reviewer 2

Summary

This paper studies a facility-location problem formulated as a submodular optimization problem. In machine learning, this problem has previously been shown to apply in the context of dataset summarization/k-medians clustering. Despite the submodularity of the objective function, solving the problem with the standard greedy algorithm is prohibitive because the complexity is k*n*m where k is the cardinality constraint and n*m is the size of the "benefit matrix". A previously suggested approach is to first sparsify the benefit matrix and then apply an optimization algorithm on the sparsified matrix. This paper refines the results from [36] by giving theoretical guarantees on the approximation ratio obtained by this method as a function of the chosen sparsification level. Two methods of sparsification are analyzed: * top t: only the top t items (most similar) are kept in each row of the benefit matrix * threshold based: only the entries of benefit matrix above the chosen threshold are kept. Finally experiments show how the suggested approach outperforms (in terms of the quality of the solution) the (non sparsified) greedy algorithm for the same running time. Or equivalently, reaching a given solution quality takes much longer if no sparsification is used.

Qualitative Assessment

The problem is interesting and well-motivated. While I haven't read paper [36] upon which this paper claims to build upon, I understand that a complete theoretical understanding of the sparsification-based approach was still missing and this paper is a good attempt at completing the picture. I think having precise theoretical guarantees is very valuable, especially since they come with almost matching lower bound (up to logarithmic factors). My only concern is about section 3.1 (threshold-based sparsification): I think this section should have a more precise discussion of the following two facts: * the stated theoretical guarantee is data dependent. Is there any way to obtain a data-independent bound under extra assumptions? For example, assume that the benefit matrix comes from a distribution, then given threshold x, the sparsification is "good" with high probability. The benefit of such a result would be to give some guidance on how to choose the threshold, or what is a good threshold (which the paper only briefly discusses). * it is also not clear how the LSH-based approach combines with the threshold-based approach. It would be nice to see an "end-to-end" theorem/proposition of the following flavor: assume the matrix is sparsified using LSH, then with high probability, we get approximation of... Finally, the experiments are interesting and convincing. However, while the conclusion of section 4.1 is clear, the results of section 4.2 are hard to interpret: there is no metric to quantify the quality of the solution and it seems that the reader is simply invited to "eyeball" the results, as a consequence I am not sure that this section is necessary.

Confidence in this Review

2-Confident (read it all; understood it all reasonably well)


Reviewer 3

Summary

This paper studies the facility location problem and analyzes sparsifications of the benefit matrix. The previous sparsification of [36] assumes the existence of a probability distribution on entries. The authors provide the improved theoretical analysis and a lower bound on sparsity level. Also, they provide a new algorithm using LSH and random walk. Numerical experiment establishes speeding up in several orders of magnitude.

Qualitative Assessment

I feel this paper is well-written and enjoyable. This paper contributes a lot in both theoretical and empirical aspects on sparsity methods. I like the numerical experiments since they compare proposed methods with other methods in detail. Overall, I think this paper provides very interesting results.

Confidence in this Review

2-Confident (read it all; understood it all reasonably well)


Reviewer 4

Summary

This paper studies submodular maximization problem based on facility location objective motivated by summarization applications. Given two sets V, I, and a benefit matrix between them, they want to pick k representatives from V to maximize the sum of maximum benefit for each element in I to the representatives in I. This matrix could be large, and make naive approaches superquadratic in computation. Prior to this work, nearest neighbor approaches have been developed to capture top benefits for each element in I and sparsify the matrix. Previous work makes strong assumptions to achieve provable guarantees. This work has stronger provable guarantees using milder assumptions, and provides subquadratic methods to summarize the data.

Qualitative Assessment

The problem is well motivated, and it makes sense to sparsify data to make existing algorithms more scalable. In general, authors propose interesting sparsification methods, and provide guarantees for a complete range of parameters without making limiting assumptions on the data. They also run large scale experiments to support their theoretical results. A more elaborate overview of analysis would have been helpful. For instance, in lines 140-141 you could elaborate on why "it cannot be much larger than..." Line 201: Did you mean keep instead of drop? Dropping 0.01 fraction does not change the size by much! Lines 236-237: Fix the typo and grammar error. Line 255: subset of

Confidence in this Review

2-Confident (read it all; understood it all reasonably well)


Reviewer 5

Summary

This paper deals with the acceleration of the function evaluation oracle for the facility location problem, a special case of the monotone submodular maximization. The authors analyze a new method, threshold sparsification, as well as an existing method by Wei et al., t-nearest neighbor. Both methods sparsify the given matrix C by replacing small value elements with zero. The approximation guarantees are shown for both methods. The one for t-nearest neighbor method needs fewer assumptions than the existing one by Wei et al. Implemented with locality sensitive hashing, the proposed method is experimentally evaluated with the summarization problem. The experimental results indicate the efficiency of the proposed method compared to the naive implementation of the sparsification method and the stochastic greedy algorithm (Mirzasoleimann et al. 2015).

Qualitative Assessment

Totally this paper gives good contribution to this topic, but it is not enough for NIPS both theoretically and practically. I weakly suggest to reject this paper. There are some places to be modified. - In experiments, the stochastic greedy is compared with the proposed method, but the stochastic greedy is the acceleration of the greedy algorithm itself, and the proposed method is the acceleration of the function evaluation oracle. I think it is inappropriate to compare them. Is it possible to combine the stochastic greedy technique with the proposed method? - The roles of alpha are different between Theorem 1 and Theorem 4. In Theorem 1, the larger alpha becomes, the worse approximation is. On the other hand, in Theorem 4, the larger alpha becomes, the better approximation is. I think it is confusing and different characters should be used. - Line 205: I cannot understand how to apply Lemma 5 in the case of C is symmetric. Please state in more formal style. - The assumptions in Proposition 10, Appendix are unclear. Please clarify which of t-nearest neighbor and threshold sparsification is used. Typos - Line 111: method method -> method - Line 211: with with -> with - Line 236: methods methods -> methods - Algorithm 1 the second row from below: parenthesis ( is small. - Appendix Line 42: np + m(1-m)^t -> np + m(1-p)^t - Appendix the third row of the equations below of Line 51: this line should be deleted. - Appendix the fifth row of the equations below of Line 51: = -> \le - Appendix the right hand side of the equations below of Line 59: f(a_1 | O \ {o_k}) -> f(o_k | O \ {o_k}) - Appendix the equations below of Line 62: the definition of i is missed. (i=1,\cdots, k) should be added. - Appendix Line 68: n \times n + 1 matrix -> n \times (n + 1) matrix - Appendix Line 105: |B|c \delta n \le c^2 \delta^2 n -> |B|c \delta n \le c^2 \delta^2 n^2 - Appendix Line 106: (1/2 - c^2) \delta^2 n -> (1/2 - c^2) \delta^2 n^2

Confidence in this Review

2-Confident (read it all; understood it all reasonably well)